# AdverMCTS: Combating Pseudo-Correctness in Code Generation via Adversarial Monte Carlo Tree Search

Qingyao Li [1]   Weiwen Liu [1]   Weinan Zhang [1]   Yong Yu [1]   Bo An [2]

## Abstract

Recent advancements in Large Language Models (LLMs) have successfully employed search-based strategies to enhance code generation. However, existing methods typically rely on static, sparse public test cases for verification, leading to *pseudo-correctness*—where solutions overfit the visible public tests but fail to generalize to hidden test cases. We argue that optimizing against a fixed, weak environment inherently limits robustness. To address this, we propose ADVERMCTS, a novel adversarial Monte Carlo Tree Search framework that combats *pseudo-correctness* by coupling code search with active vulnerability discovery. ADVERMCTS formulates generation as a minimax-style game between a *Solver* agent, which synthesizes code candidates, and an *Attacker* agent, which evolves to generate targeted corner test cases that exploit logical divergences in the current code pool. These discovered tests form a dynamic, progressively hostile filter that penalizes fragile reasoning. Extensive experiments demonstrate that ADVERMCTS significantly outperforms state-of-the-art baselines, effectively reducing false positive rates and forcing the model to generalize beyond the initial constraints. The resources of this work are available at https://github.com/SIMONLQY/AdverMCTS.

## 1. Introduction

The advent of Large Language Models (LLMs) has fundamentally transformed the landscape of automated code generation (Odeh et al., 2024; Jiang et al., 2024a; Yang et al., 2025b; Dong et al., 2025b), enabling systems to solve stan-

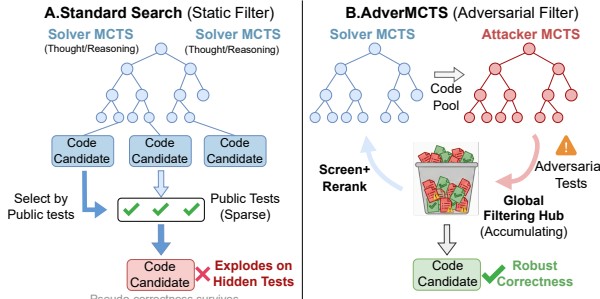

*Figure 1.* Conceptual Comparison. (A) Standard Search relies on sparse public tests, creating a "leaky" filter prone to pseudo-correctness. (B) ADVERMCTS employs an active Attacker to co-evolve a progressively stricter environment, exposing hidden bugs and enforcing robust correctness.

dard programming tasks with remarkable proficiency (Chen, 2021; Li et al., 2022; Wang et al., 2025). However, as the focus shifts towards complex, competition-level problems that demand deep algorithmic reasoning, the paradigm is evolving from simple "System 1" token prediction to "System 2" deliberation (Xiang et al., 2025; Li et al., 2025f), often characterized as Test-Time Compute (TTC) scaling (Brown et al., 2024; Snell et al., 2024). Central to this shift is the reliance on execution-based verification (Zhong et al., 2024; Ni et al., 2023; Dong et al., 2025a), where the correctness of a generated code is inferred from its behavior on a set of test cases. In this context, the interplay between code synthesis and test validation has become a pivotal axis for advancing model performance.

To leverage this verification signal, recent research has introduced various search-guided frameworks that structure decoding as a navigation problem (Chen et al., 2024a; Gou et al., 2024; Lyu et al., 2025). Prominent approaches such as Tree of Thoughts (ToT) (Yao et al., 2023) and Language Agent Tree Search (LATS) (Zhou et al., 2023) integrate planning with Breadth First Search (Kurant et al., 2010) and Monte Carlo Tree Search (MCTS) (Browne et al., 2012), using the LLM as both a policy and a value estimator. Similarly, methods like PG-TD (Zhang et al., 2023) and CodeT (Chen et al., 2022) utilize rollout execution on sample tests to guide the generation trajectory. These search-based strategies have successfully pushed the boundaries of code generation, achieving significant performance gains on bench-

[1]Shanghai Jiao Tong University, Shanghai, China [2]Nanyang Technological University, Singapore. Correspondence to: Weinan Zhang <wnzhang@sjtu.edu.cn>, Bo An <boan@ntu.edu.sg>.

*Proceedings of the 43rd International Conference on Machine Learning*, Seoul, South Korea. PMLR 306, 2026. Copyright 2026 by the author(s).

marks by enabling lookahead and backtracking capabilities that are absent in greedy decoding (Li et al., 2025b).

Despite these advancements, we argue that a key challenge is still under-addressed: *pseudo-correctness*—generating solutions that overfit the public tests while failing on the underlying logic required by the hidden test suite. Public tests typically sample from simplified sanity checks, whereas hidden tests probe the long-tail of corner cases. Consequently, static verification signals are frequently insufficient to expose hidden bugs, and even sophisticated search can be misled into preferring fragile code, creating a *survivorship bias* where many "surviving" candidates are merely overfitted solutions. We empirically verify this bottleneck in Appendix C.1, confirming that while search spaces often contain correct solutions, sparse public tests fail to identify them. Thus, the bottleneck is not the Solver's capacity to generate correct solutions, but the environment's capacity to *discriminate* them at inference time. This calls for a mechanism that actively strengthens verification, rather than merely expanding the candidate set.

This paper presents a unique perspective on the problem: robust code generation should be viewed as an adversarial game between a solver that proposes solutions and an attacker that actively searches for failure-inducing tests. Intuitively, if the search procedure is only navigating within a fixed, weak environment (the public tests), then it is optimizing the wrong objective. What we need at test time is a mechanism that co-evolves solutions and constraints: as the solver improves, the environment should become more hostile, continuously surfacing new corner cases that invalidate pseudo-correct codes. Figure 1 illustrates this paradigm shift: while standard search is limited by static verification, our approach constructs a dynamic hostile environment to mitigate survivorship bias.

Building on this perspective, we propose ADVERMCTS, an adversarial Monte Carlo tree search framework that addresses pseudo-correctness by coupling solution search with vulnerability discovery. ADVERMCTS instantiates a minimax-style interaction between two agents: (i) a Solver MCTS that searches over chain-of-thought trajectories and continuously synthesizes executable candidate codes during simulation, and (ii) an Attacker MCTS that conditions on the evolving code pool and searches for discriminative tests via divergence-driven test synthesis. A global test filter adjudicates and accumulates valid corner cases into a dynamic test suite, which is then used as hard constraints for screening and final re-ranking codes—forcing the solver to generalize beyond the initial public tests. Our contributions are summarized as follows:

- We propose ADVERMCTS, a dual-agent search framework that enhances code robustness through iterative adversarial interaction. To the best of our knowledge, AD-

VERMCTS is the first to unify code search with an active adversarial test search at test time to tackle competition-level programming problems.

- We develop key mechanisms that make adversarial test-time search effective in practice, including a persistent attacker tree, divergence-driven test synthesis, and a global test filter that turns discovered corner cases into reusable hard constraints.

- Extensive experiments demonstrate that ADVERMCTS significantly outperforms state-of-the-art baselines. Notably, our analysis reveals that the adversarial pressure effectively reduces the false positive rate of generated codes, validating the efficacy of "hostile" supervision in test-sparse environments.

## 2. Related Work

**Competition-level Code Generation** Early LLMs for code such as OpenAI Codex (Chen, 2021) achieved strong results on standard benchmarks (Zhou et al., 2023), yet remained challenged by competition-level problems that require deeper algorithmic reasoning (Li et al., 2023a; Lozhkov et al., 2024; Hui et al., 2024; Paul et al., 2024). AlphaCode (Li et al., 2022) marked the first non-trivial breakthrough in competitive programming, reaching median competitor-level performance on Codeforces (Mirzayanov et al., 2020). Building on this, a growing line of work improves reliability via search-guided generation (Princis et al., 2025; Jiang et al., 2024b; Wang et al., 2024; Chen et al., 2024c; Gao et al., 2024; Li et al., 2025c;e). Representative examples include PG-TD (Zhang et al., 2023), which executes candidate programs on sample tests for lookahead planning; CodeT (Chen et al., 2022), which generates additional tests to select solutions and notably improves pass@1 on HumanEval (Li & Murr, 2024); and LATS (Zhou et al., 2023), which integrates MCTS into code generation with the LLM as the policy. While these methods allocate computation to sampling or single-agent search over code, our approach instead introduces an adversarial MCTS framework that couples a code-generating Solver with a test-generating Attacker, enabling targeted failure discovery and more robust selection beyond prior single-agent search paradigms.

**Test Time Computing Scaling** Test-Time Computing (TTC) frames inference-time compute as a key lever for improving model performance (Muennighoff et al., 2025; Chen et al., 2024b; Zhang et al., 2025; Li et al., 2025a; Zeng et al., 2025), enabling deeper "System 2" style reasoning (Brown et al., 2024; Snell et al., 2024). This motivates search-structured decoding, where methods such as Tree of Thoughts (ToT) (Yao et al., 2023) and Reasoning via Planning (RAP) (Hao et al., 2023) extend Chain-of-Thought (Wei et al., 2022) with lookahead/backtracking, and RethinkMCTS (Li et al., 2025e) further repairs erro-

neous nodes using fine-grained feedback. However, these approaches largely verify against static signals (e.g., LATS or RethinkMCTS), similar in spirit to fixed-signal self-improvement such as Reflexion (Shinn et al., 2023), and differ from adversarial training efforts like ATGen (Li et al., 2025d). ADVERMCTS instead introduces an active adversary at test time: an Attacker that adaptively hardens the verification environment, forcing the Solver to generalize beyond the initial constraints.

## 3. ADVERMCTS

### 3.1. Problem Formulation

Given a natural language problem description $P$, robust code generation aims to synthesize a program $C$ that satisfies the underlying semantics of $P$. Correctness is evaluated by executing $C$ on a test suite $\mathcal{T}$, which is split into a small set of public tests $\mathcal{T}_{\text{pub}}$ and a larger hidden set $\mathcal{T}_{\text{hidden}}$. At inference time, the solver observes only $(P, \mathcal{T}_{\text{pub}})$, where $\mathcal{T}_{\text{pub}}$ provides limited verification and may fail to expose corner cases. Consequently, passing $\mathcal{T}_{\text{pub}}$ does not guarantee generalization to $\mathcal{T}_{\text{hidden}}$. Our goal is to learn a policy that, using only $(P, \mathcal{T}_{\text{pub}})$, produces a solution $C^*$ that maximizes performance on $\mathcal{T}_{\text{hidden}}$.

### 3.2. Overview

We propose ADVERMCTS, an adversarial MCTS framework to enhance code generation robustness. As illustrated in Figure 2, the framework operates as an iterative game between two agents: a *Solver* and an *Attacker*. The *Solver* aims to synthesize robust code that satisfies the problem requirements, while the *Attacker* actively searches for "vulnerability" test cases that induce behavioral divergence in the *Solver*'s generated code. The two agents interact through a shared *Code Pool* and a dynamic *Global Test Filter*, iteratively refining both the solution and test quality. The detailed pseudocode is provided in Algorithm 1 in the Appendix E.

### 3.3. Solver MCTS: Robust Code Generation

The Solver aims to synthesize a robust program $C$ given the problem description $P$. The search tree acts as a structured reasoning space, where the root represents the initial state $s_0 = P$, and each node $s_t = \{P, \tau_1, \ldots, \tau_t\}$ represents a partial solution consisting of a sequence of Chain-of-Thought (CoT) steps. The Solver iteratively builds the search tree through four phases: *Selection*, *Expansion*, *Simulation*, and *Backpropagation*.

**Selection.** In each iteration, the algorithm traverses the tree from the root to a leaf node by recursively selecting the child node that maximizes the Upper Confidence Bound (UCB) (Silver et al., 2017). Formally, for a parent node

$s$ and a child node $s'$ (reached by action $a$), the selection policy is defined as:

$$a^* = \operatorname*{argmax}_a \left( Q(s,a) + c_{\text{puct}} \cdot P(s,a) \frac{\sqrt{N(s)}}{1 + N(s,a)} \right), \tag{1}$$

where $Q(s,a)$ is the estimated value of the action, $P(s,a)$ is the prior probability given by the LLM, $N(s)$ is the visit count of the parent node, and $c_{\text{puct}}$ is the exploration constant. This mechanism balances the exploitation of high-quality reasoning paths with the exploration of uncertain branches.

**Expansion.** Once a leaf node $s_{\text{leaf}}$ is reached, the Solver expands it by sampling $k$ potential next-step thoughts $\{\tau^{(1)}, \ldots, \tau^{(k)}\}$ from the LLM policy $\pi_\theta(\cdot|s_{\text{leaf}})$. These thoughts typically represent intermediate reasoning steps or algorithm designs. Each new thought $\tau$ creates a new child node appended to the current trajectory.

**Simulation and Code Synthesis.** Standard MCTS typically performs random rollouts to estimate value. In our context, however, we leverage the LLM's completion capability to perform a *semantic simulation*. For a newly expanded thought node $s_{\text{new}}$, the Solver performs a rollout to generate a complete code candidate $C$. This design ensures that valid code candidates are generated continuously throughout the search process, rather than only at the search depth limit. Consequently, generated candidates are subjected to immediate screening: only those that pass both the public tests $\mathcal{T}_{\text{pub}}$ and previous accumulated adversarial suite in the *Global Test Filter* $\mathcal{T}_{\text{global}}$ are deposited into the shared *Code Pool* to support the Attacker's subsequent operations.

**Backpropagation with Hybrid Feedback.** After simulation, the generated code $C$ is evaluated to compute a reward $R$. The reward is a composite signal from two sources:

- **Intrinsic Correctness ($R_{\text{pub}}$):** The pass rate on the visible public test cases $\mathcal{T}_{\text{pub}}$.

- **Extrinsic Interaction ($V_{\text{penalty}}$):** A penalty signal fed back from the Global Test Filter (see Figure 2, blue arrow "Backpropagation(-V)"). If the generated code $C$ is later found to fail on the global adversarial tests $\mathcal{T}_{\text{global}}$ or the newly generated $T_{\text{new}}$, a penalty $-V$ is applied to the current node of the solver.

The final value estimate $Q(s,a)$ of all nodes along the trajectory is updated using the collected reward, penalizing reasoning paths that lead to fragile code and reinforcing those that survive the adversarial filtering.

### 3.4. Attacker MCTS: Vulnerability Discovery

Parallel to the Solver, the Attacker operates to expose defects in the generated codes. Unlike standard test generation,

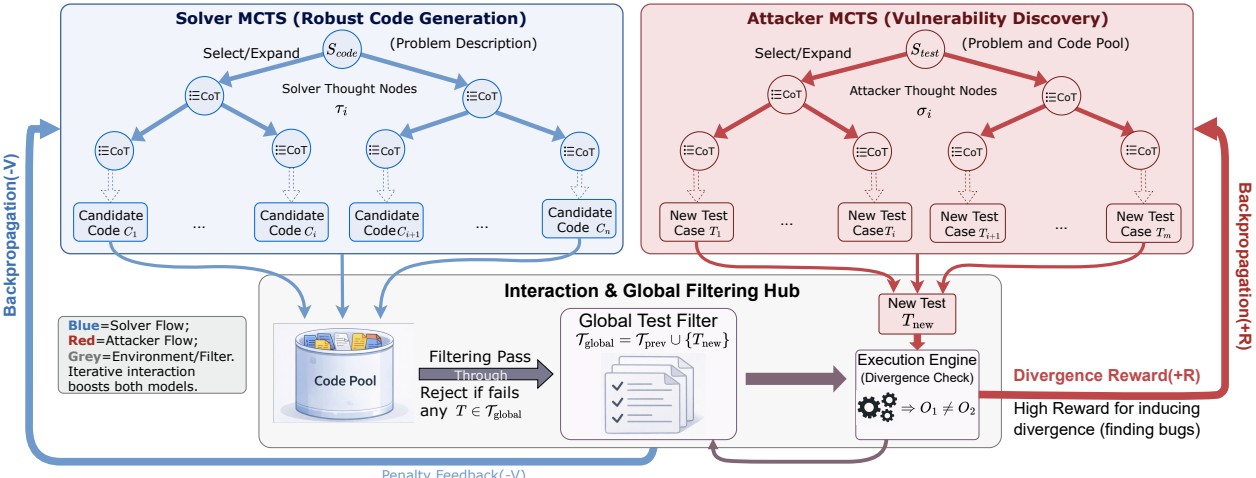

*Figure 2.* Overview of ADVERMCTS. A minimax interaction where the Solver (blue) generates code and the Attacker (red) synthesizes adversarial tests. The Global Hub turns valid attacks into constraints. Feedback is dual: divergence rewards (+R) for the Attacker and penalties (-V) for the Solver enforce robust generalization.

which blindly targets the problem description, our Attacker employs a targeted adversarial strategy conditioned on the evolving *Code Pool*.

**Persistent Search with an Evolving Code Pool.** The Attacker maintains a persistent search tree that grows incrementally throughout the searching process. The root state represents the problem context $S_{\text{test}} = \{P, \mathcal{C}_{\text{pool}}\}$, where $\mathcal{C}_{\text{pool}}$ is the dynamic set of currently accepted code that have passed all public tests. Crucially, this tree is *retained* across iterations. When $\mathcal{C}_{\text{pool}}$ reaches a certain capacity, the Attacker resumes search from the existing tree structure, allowing it to progressively refine its attack strategies against an increasingly robust population of codes.

**Selection and Expansion.** The selection phase mirrors the Solver's logic, using a UCB-based policy to navigate to the most promising attacker thought nodes $\sigma$. At the frontier, the Attacker expands a new thought node $\sigma_{\text{new}}$ representing a specific testing strategy (e.g., "Check boundary condition for $N = 0$" or "Test with large prime inputs"). This reasoning step guides the subsequent generation towards specific vulnerability types rather than random fuzzing.

**Simulation: Divergence-driven Test Synthesis.** Upon reaching a new thought node $\sigma_{\text{new}}$, the Attacker employs a Divergence-driven Multi-sample Test Synthesis (DMTS) strategy to ensure high-quality test generation. The LLM generates a batch of $k$ candidate test inputs $\{T^{(1)}, \ldots, T^{(k)}\}$ conditioned on the current code pool: $\{T^{(1)}, \ldots, T^{(k)}\} \sim \pi_{\text{adv}}(\cdot | P, \mathcal{C}_{\text{pool}}, \sigma_{\text{new}})$.

By explicitly conditioning on $\mathcal{C}_{\text{pool}}$, the Attacker aims to craft inputs that exploit the logical discrepancies observed among the current code. Generating multiple candidates reduces the variance of the generation process and increases

the probability of discovering a valid corner case.

Since the code pool consists of public-test–passing candidates and no ground-truth output is available at test time, we reward inputs that expose pseudo-correctness by causing the candidates in the pool to produce diverging outputs. With this principle, we propose to evaluate the *Divergence Reward* $R_{div}$ for each candidate. The reward is defined based on the disagreement of outputs produced by the Code Pool:

$$R_{\text{div}}(T) = \mathbb{I}\left[\exists i, j : O_i(T) \neq O_j(T)\right] \cdot w_{\text{div}}, \quad (2)$$

where $O_i(T)$ denotes the execution output of code $C_i \in \mathcal{C}_{\text{pool}}$ on input $T$; $w_{\text{div}}$ is to normalize this reward by the number of unique outputs. We select the most discriminative test case $T^* = \text{argmax}_{\text{m}} R_{\text{div}}(T^{(\text{m})})$ from the batch.

While output divergence cannot detect the case where all candidates agree on the same wrong output, it is aligned with our test-time objective: distinguishing pseudo-correct code that already pass $\mathcal{T}_{\text{pub}}$. In practice, disagreement provides a high-signal trigger for discovering latent logical gaps, and we further guard against invalid via the Arbiter-based validity check.

**Backpropagation.** The reward $R_{\text{div}}(T^*)$ is backpropagated up the Attacker's tree, reinforcing the reasoning strategies that lead to high-divergence scenarios.

### 3.5. Interaction and Global Filtering Hub

The interaction module acts as the central "arena" where the Solver and Attacker continuously exchange feedback. This mechanism ensures that only valid, high-quality test cases are retained to penalize fragile code.

**LLM-based Output Arbiter.** When the Attacker generates a test case $T_{\text{new}}$ that induces output divergence among

the code candidates (i.e., $\exists C_i, C_j : O_i(T_{\text{new}}) \neq O_j(T_{\text{new}})$), a ground truth label is required to determine which candidate is at fault. Since hidden tests are unavailable during inference, we introduce an *LLM-based Output Arbiter* to adjudicate the results. The Arbiter takes the problem description $P$, the generated input $T_{\text{new}}$, and the divergent outputs $\{O_i, O_j, \dots\}$ as input. It analyzes the semantic logic of $P$ to identify the correct expected output $O^*$.

$$O^*, \text{validity} \leftarrow \text{Arbiter}(P, T_{\text{new}}, \{O_i, O_j\}). \quad (3)$$

If the Arbiter deems the test input $T_{\text{new}}$ ambiguous or invalid, the test is discarded. Otherwise, the pair $(T_{\text{new}}, O^*)$ is formalized as a new valid test case.

**Global Filtering and Feedback Loops.** Successfully adjudicated tests are added to the *Global Test Filter* ($\mathcal{T}_{\text{global}}$), a dynamic repository of "hard" corner cases found during the search.

- **Penalty Feedback (-V) for Solver:** With the adjudicated ground truth $O^*$, the system identifies the specific candidates $\mathcal{C}_{\text{fail}}$ that produced incorrect outputs. A penalty value $-V$ is backpropagated to their corresponding thought nodes in the Solver's MCTS tree. This signal discourages the Solver from pursuing reasoning paths that led to these fragile implementations.

- **Filtering Mechanism:** The $\mathcal{T}_{\text{global}}$ acts as a gatekeeper. In subsequent iterations, any new candidate generated by the Solver must pass all tests in $\mathcal{T}_{\text{global}}$ before entering the Code Pool. This ensures that the Code Pool monotonically improves in robustness.

### 3.6. Inference: Robust Selection

After the search process, the Solver yields a diverse set of candidate programs. To select the final submission $C^*$, we employ a hierarchical *Test-based Reranking* strategy that prioritizes robustness against the accumulated adversarial knowledge.

The code candidates are ranked by a two-stage criterion:

- **Primary Sort (Public Integrity):** Candidates are first ranked by their pass rate on the public test suite $\mathcal{T}_{\text{pub}}$. This step filters out solutions that fail to meet the basic problem requirements.

- **Secondary Sort (Adversarial Robustness):** For candidates with identical public test scores (which is common due to the scarcity of $\mathcal{T}_{\text{pub}}$), we further rank them by their pass rate on the generated adversarial tests in $\mathcal{T}_{\text{global}}$.

This mechanism effectively resolves ties among "pseudo-correct" solutions that overfit the public tests, selecting the candidate that survives the hostile environment constructed by the Attacker.

## 4. Experiments

### 4.1. Experiment Settings

**Datasets.** We evaluate ADVERMCTS on two challenging competition-level code generation benchmarks: APPS (Hendrycks et al., 2021) and TACO (Li et al., 2023b). The APPS dataset contains three levels of difficulties: *Introductory*, *Interview*, and *Competition*. And *Easy*, *Medium*, and *Hard* split for TACO. Followin prior work (Li et al., 2025e), we evaluate all the methods on the formal 100 problems per split. For each problem, we set the maximum number of $|\mathcal{T}_{\text{pub}}| = 5$. Following prior work (Austin et al., 2021; Chen, 2021; Dong et al., 2025a), we use the standard *pass rate* and *pass@1* metrics evaluated on the hidden test suite to measure robust correctness.

**Baselines.** We compare ADVERMCTS against state-of-the-art methods spanning three categories: (1) Direct Synthesis: Standard *Zero-shot* prompting and *Best-of-N* sampling (with $N = 16$) to establish performance lower bounds. (2) Search and Planning: Advanced search-based frameworks including PG-TD (Zhang et al., 2023), Tree of Thoughts (ToT) (Yao et al., 2023), and LATS (Zhou et al., 2023), which utilize lookahead planning or self-reflection. (3) MCTS Variants: We also compare with MCTS-Thought (MCTS for reasoning search) and RethinkMCTS (Li et al., 2025e), a recent method focusing on repairing erroneous nodes. Due to space constraints, detailed descriptions of these baselines and their specific configurations are provided in Appendix B.

**Implementation Details.** We employ the Qwen3-4B-Instruct-2507 and Qwen3-8B (non-instruct) (Yang et al., 2025a) as the main backbone LLM for both the Solver and Attacker agents. We also experiment on DeepSeek-V3.2 (Liu et al., 2025) in the scaling experiment. For the Solver, we implement a standard MCTS with a UCB exploration constant $c_{puct} = 4$. For the Attacker, we set a moderate search budget of $N = 2$ rollouts per iteration, balancing computational efficiency with adversarial strength. The maximum interaction depth is set to 16 rollouts and penalty value $V = 0.1$. All experiments are conducted using the vLLM library (Kwon et al., 2023) for efficient inference.

### 4.2. Main Results

Table 1 reports results on APPS and TACO with two backbones. Overall, ADVERMCTS achieves the best performance across benchmarks and backbones, with especially strong gains on APPS and on the TACO average. Comparing baselines, we observe a clear granularity effect: token-level lookahead (PG-TD) offers limited gains due to its myopic horizon; code-level search (LATS) improves performance

*Table 1.* Main Results. We report Pass Rate and Pass@1 accuracy using Qwen3-4B-Instruct/8B backbones under rollout 16. ADVERMCTS consistently outperforms state-of-the-art search baselines, showing significant gains on the most challenging subsets.

| | APPS | | | | | | | | TACO | | | | | | | |
| Model | Pass Rate (%) | | | | Pass@1 (%) | | | | Pass Rate (%) | | | | Pass@1 (%) | | | |
| | Intro. | Inter. | Comp. | Avg. | Intro. | Inter. | Comp. | Avg. | Easy | Medium | Hard | Avg. | Easy | Medium | Hard | Avg. |
| **Qwen3-4B-Instruct** | | | | | | | | | | | | | | | | |
| Base | 51.94 | 53.30 | 31.70 | 45.64 | 35 | 27 | 15 | 25.67 | 57.27 | 39.97 | 29.87 | 42.37 | 33 | 20 | 10 | 21.00 |
| Base (16) | 65.27 | 66.52 | 48.77 | 60.18 | 46 | 40 | 23 | 36.33 | 68.65 | 56.15 | 48.49 | 57.76 | 37 | 24 | 12 | 24.33 |
| PG-TD | 71.88 | 69.05 | 40.50 | 60.48 | 56 | 43 | 27 | 42.00 | 70.99 | 61.81 | 56.54 | 63.11 | 50 | 35 | 20 | 35.00 |
| MCTS-Thought | 75.56 | 74.37 | 58.10 | 69.34 | 57 | 44 | 29 | 43.44 | 82.13 | 68.10 | 65.83 | 72.02 | 47 | 32 | 14 | 31.00 |
| RethinkMCTS | 77.77 | 72.69 | 61.13 | 70.53 | 56 | 47 | 29 | 43.67 | 83.91 | 68.79 | 64.75 | 72.48 | 52 | 31 | 18 | 33.67 |
| ToT | 65.63 | 66.46 | 46.83 | 59.64 | 46 | 43 | 33 | 40.67 | 77.47 | 64.43 | 56.98 | 66.29 | 52 | 34 | 14 | 34.00 |
| LATS | 71.88 | 68.58 | 53.25 | 64.57 | 50 | 39 | 27 | 38.67 | 75.46 | 63.94 | 58.34 | 65.91 | 44 | 29 | 17 | 30.00 |
| ADVERMCTS | **78.63** | **75.24** | **64.45** | **72.77** | **61** | **52** | **36** | **49.67** | **84.04** | **73.05** | **66.23** | **74.44** | **55** | **37** | **22** | **38.00** |
| **Qwen3-8B** | | | | | | | | | | | | | | | | |
| Base | 42.97 | 40.21 | 25.95 | 36.38 | 21 | 17 | 7 | 15.00 | 45.06 | 29.08 | 24.97 | 33.04 | 22 | 9 | 4 | 11.67 |
| Base (16) | 50.18 | 46.75 | 38.87 | 45.27 | 26 | 21 | 15 | 20.67 | 51.13 | 35.19 | 34.97 | 40.43 | 23 | 8 | 7 | 12.67 |
| PG-TD | 56.08 | 53.81 | 32.67 | 47.52 | 30 | 29 | 14 | 24.33 | 63.59 | 46.11 | 44.09 | 44.09 | 39 | 18 | 13 | 23.33 |
| MCTS-Thought | 62.80 | 60.55 | 50.40 | 57.92 | 37 | 36 | 21 | 31.33 | 70.15 | 56.11 | 49.69 | 58.65 | 36 | 21 | 8 | 21.67 |
| RethinkMCTS | 66.68 | 62.16 | 52.18 | 60.34 | 37 | 35 | 23 | 31.67 | 76.35 | 60.82 | 48.89 | 62.02 | 42 | 22 | 12 | 25.33 |
| ToT | 50.52 | 49.22 | 33.67 | 44.47 | 28 | 22 | 16 | 22.00 | 66.49 | 46.73 | 43.43 | 52.21 | 39 | 21 | 13 | 24.33 |
| LATS | 59.64 | 56.01 | 42.30 | 52.65 | 30 | 31 | 17 | 26.00 | 62.52 | 51.08 | 38.03 | 50.54 | 32 | 21 | 7 | 20.00 |
| ADVERMCTS | **66.96** | **63.15** | **53.27** | **61.13** | **44** | **40** | **24** | **36.00** | **76.99** | **60.84** | **52.95** | **63.94** | **44** | **27** | **13** | **28.00** |

but struggles with the sparsity of the full program space; whereas thought-level search (MCTS-Thought) proves most effective by decomposing complex logic into manageable planning steps. Crucially, unlike these methods that primarily allocate compute to expanding candidate volume, ADVERMCTS strategically directs budget toward *adversarial verification*. This active purification mechanism filters out pseudo-correct solutions, enabling a significantly higher conversion of public-test pass rates into robust hidden-test correctness (Pass@1).

### 4.3. Scaling with Backbone Capability

To verify the scalability and model-agnosticism of AD-VERMCTS, we extended our evaluation to the state-of-the-art frontier model, DeepSeek-V3.2 (671B) (Liu et al., 2025). As shown in Figure 3, we order the models on the x-axis by their baseline sampling performance to visualize the correlation between intrinsic model capability and method gain. The results reveal a strictly monotonic upward trend: AD-VERMCTS consistently outperforms both the strong sampling baseline (Base-16) and the search baseline (MCTS-Thought) across all settings. Notably, the performance gain remains robust even on the 671B frontier model, confirming that our adversarial mechanism is agnostic to the underlying model capacity and effectively scales from smaller open weights to massive state-of-the-art base models.

### 4.4. Ablation Study

We ablate three core components of ADVERMCTS, removing exactly one module at a time while keeping all other settings fixed: (i) *AT-Tree*, which removes the entire Attacker agent and adversarial testing, reverting to standard search guided only by public tests; (ii) *DMTS*, which disables the multi-sample test synthesis, relying on a single stochastic

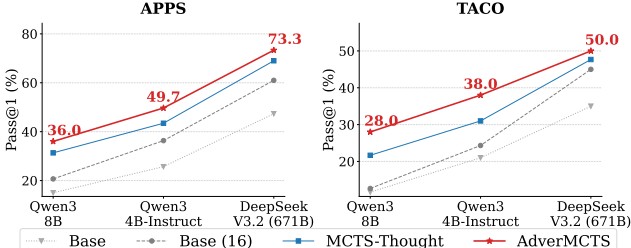

*Figure 3.* Scalability with Model Capabilities. We compare methods across backbones sorted by intrinsic capability. Despite parameter discrepancies, ADVERMCTS consistently amplifies performance, maintaining a significant lead across all model scales.

sample for test generation instead of divergence-driven selection; and (iii) *GF-Hub*, where we specifically disable *pre-admission* global screening before a candidate enters the code pool, while keeping the final test-based re-ranking unchanged.

As shown in Figure 4, all modules contribute positively. Removing *AT-Tree* yields the largest drop, since no adversarial tests are produced and the solver degenerates to verification on public tests only, making pseudo-correct candidates much harder to eliminate. Disabling *DMTS* also consistently hurts performance, indicating that multi-sample synthesis is important to reduce variance and reliably surface divergence-triggering inputs. Finally, turning off the *GF-Hub* screening step further degrades results, especially on harder splits, suggesting that early global screening helps prevent fragile solutions from polluting the pool and weakening subsequent adversarial refinement and selection.

### 4.5. Extended Analysis

**Cost-Performance Trade-off under Test-Time Compute Scaling.** We investigate the scaling efficiency of AD-VERMCTS by analyzing the Pareto frontier (Lotov & Mi-

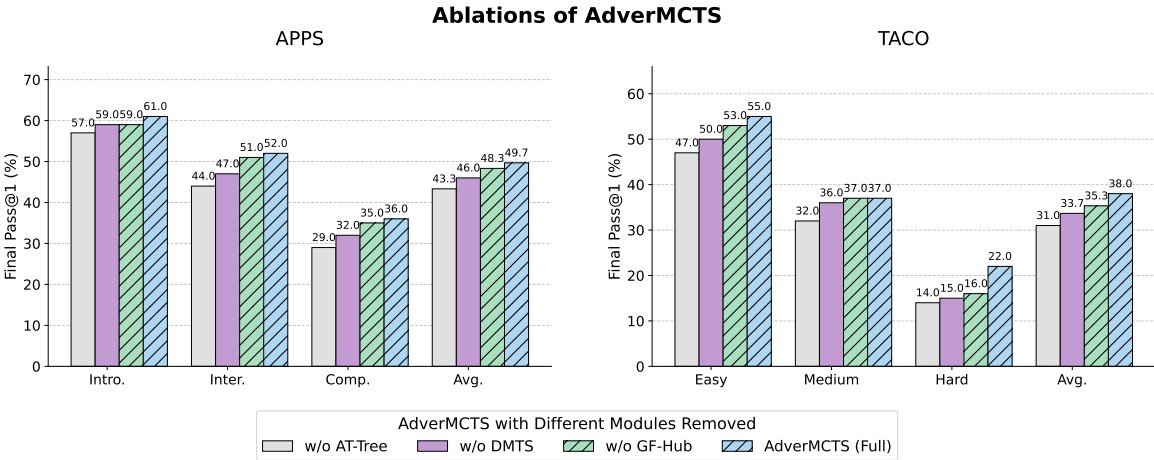

Figure 4. Ablation study of ADVERMCTS on APPS and TACO. We report the final Pass@1 (%) under the same inference budget while removing one component at a time.

ettinen, 2008) between solution correctness (Pass@1) and computational cost (average token consumption). As shown in Figure 5, across both APPS and TACO, ADVERMCTS consistently dominates the baseline frontier: for a similar token budget, it achieves higher Pass@1, indicating that the extra compute is spent on more effective verification rather than producing more redundant samples. Notably, on both datasets, ADVERMCTS with a budget of $N = 16$ rollouts achieves superior accuracy compared to the baseline with $N = 32$ rollouts, while consuming significantly fewer tokens. This observation creates a compelling counter-argument to the concern of overhead introduced by the dual-agent architecture: the active "purification" of the search space via adversarial counter-examples proves to be more compute-efficient than brute-force scaling of reasoning paths. By filtering out pseudo-correct solutions early, ADVERMCTS achieves "smarter" test-time scaling rather than simply "harder" scaling.

selected 100 distinct problems from the hardest subsets (APPS-Competition and TACO-Hard) and tracked two key metrics over 16 rollouts: 1) The Cumulative Number of Valid Adversarial Test Corner Cases (that induce divergence among codes) discovered by the Attacker (Orange); and 2) The Hidden Pass Rate of the Solver's code pool (Blue), which serves as a proxy for true semantic correctness.

Figure 6 presents the dual-axis trajectories. We observe a strong positive correlation between the accumulation of adversarial constraints and the improvement of solution robustness. Initially (Rollout 0-3), the code pool is fragile, passing a few hidden tests. As the Attacker actively identifies valid corner cases (indicated by the steep rise in the orange curve), the filtering mechanism filters out pseudo-correct codes. By Rollout 16, the accumulation of diverse test cases acts as a comprehensive logical boundary, guiding the Solver's pass rate to converge at a significantly higher level. This confirms that ADVERMCTS functions as a constructive adversarial process: adversarial test cases evolve into strict constraints, effectively pruning fragile solutions and forcing the Solver to generalize beyond the public tests.

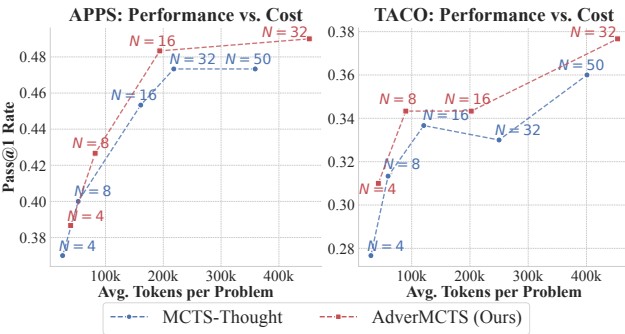

Figure 5. Cost-Performance Pareto Frontiers (upper left is better). Comparison of Pass@1 accuracy against average token consumption per problem. ADVERMCTS consistently achieves a superior Pareto frontier. The labels denote the rollout budget.

**Co-evolution of Test and Code.** To uncover the temporal mechanism behind ADVERMCTS, we visualize the iterative interaction between the Attacker and the Solver. We

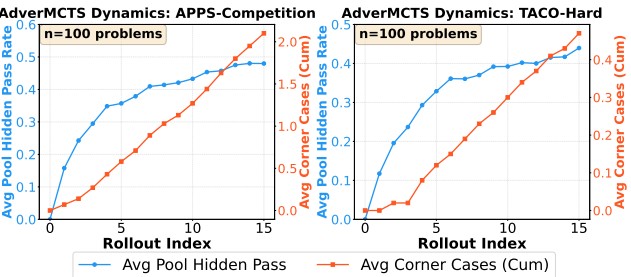

Figure 6. Visualization of Adversarial Dynamics. We track the co-evolution process. The synchronous rise of cumulative corner cases (Orange) and the Solver's hidden pass rate (Blue) demonstrates that accumulating adversarial constraints drives the Solver toward robust solutions.

**Discriminative Power of Adversarial Test Generation.**

To show that the Attacker provides a *meaningful* supervision signal (rather than stochastic noise), we evaluate how well its generated adversarial tests discriminate *Pseudo-Correct* solutions (pass $\mathcal{T}_{pub}$ but fail $\mathcal{T}_{hidden}$) from *True Correct* ones (pass all tests). Figure 7 visualizes the impact of the generated adversarial tests on these two groups. The results highlight three key observations: (1) **High Sensitivity:** It achieves a Recall of 56.7% (165/291), exposing over half of the "silent bugs" missed by standard MCTS; (2) **Conservative Behavior:** It maintains a low False Positive Rate of 16.1%, indicating the generated corner cases are largely valid; and (3) **High Precision:** The overall precision of 78.95% confirms that output divergence is a reliable proxy for semantic correctness without ground truth.

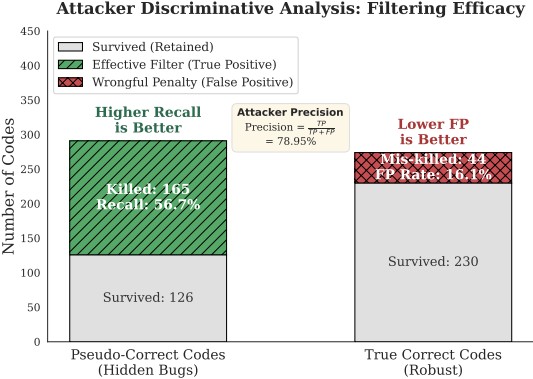

*Figure 7.* Attacker Discriminative Analysis. We evaluate filtering efficacy on *Pseudo-Correct* (pass public, fail hidden) versus *True Correct* codes. The **green** hatched area marks effective bug identification (True Positives), while the **red** area indicates wrongful penalization (False Positives).

**Solver-Aware is Crucial for Effective Attacker.** We investigate two critical design choices to see the impact of solver awareness for the attacker: (1) Solver Context, by ablating the Attacker's access to the code pool $\mathcal{C}_{pool}$ ("w/o Solver Context"); and (2) Search Algorithm, we examine whether this solver-to-attacker signal can be *effectively exploited* by different search strategies, by replacing MCTS with lighter Best-of-N baseline and Random-MCTS baseline where the *UCB-based selection* in MCTS is replaced by random selection.

Table 2 yields two key insights. First, removing solver context significantly degrades performance. Without observing $\mathcal{C}_{pool}$, the Attacker generates generic test cases rather than targeted counter-examples that exploit the Solver's specific logical gaps. Second, MCTS outperforms Best-of-N and Random-MCTS not simply by searching "harder", but because its selection, value estimation, and backpropagation can absorb solver-derived feedback. This allows the Attacker to refine test reasoning progressively, whereas one-shot sampling is limited to the immediate quality of samples and cannot reliably leverage the feedback.

*Table 2.* Ablation on Attacker Optimization. We analyze the impact of *Solver Context* and *Search Strategy*. The results validate that both contextual targeting and iterative tree search are essential for finding meaningful vulnerabilities.

| Method | APPS (Pass@1) | | | | TACO (Pass@1) | | | |
|---|---|---|---|---|---|---|---|---|
| | Intro. | Inter. | Comp. | Avg. | Easy | Med. | Hard | Avg. |
| **AdverMCTS (Full)** | **61.0** | **52.0** | **36.0** | **49.7** | **55.0** | **37.0** | **22.0** | **38.0** |
| *Information Cutoff* | | | | | | | | |
| w/o Solver Context | 58.0 | 49.0 | 31.0 | 46.0 | 52.0 | 35.0 | 17.0 | 34.7 |
| *Search Strategy* | | | | | | | | |
| Best-of-N | 58.0 | 48.0 | 31.0 | 45.7 | 53.0 | 31.0 | 18.0 | 34.0 |
| Random-MCTS | 57.0 | 47.0 | 27.0 | 43.7 | 47.0 | 34.0 | 16.0 | 32.3 |

**Output Judging Matters: LLM Arbiter vs. Majority Voting.** In ADVERMCTS, accurately determining the expected output for a generated test input is critical. Once a test input induces disagreement among codes in the pool, we must determine which output is correct to label the test and update the global test filter. We compare two judging strategies: (*i*) *Majority Voting*, which selects the majority output among the current code pool, and (*ii*) our *LLM Arbiter*, where the same backbone LLM adjudicates the correct output by reading the problem specification and the candidate outputs. As shown in Table 3, the Arbiter significantly outperforms Voting on both benchmarks. Crucially, on the TACO dataset, we analyzed the validity of the test pairs generated by both methods. The Voting strategy resulted in a low rate of valid test pairs, indicating that the majority of solutions frequently converged on incorrect outputs. These results justify the design choice of using an LLM arbiter: it improves both the quality of adversarial labels and the effectiveness of adversarial verification.

*Table 3.* Comparing output-judging strategies for adversarial tests. The LLM arbiter yields higher downstream performance and substantially higher labeled-test validity(measured against gold code).

| Oracle Strategy | Signal Quality (TACO) | Performance (Pass@1) | |
|---|---|---|---|
| | Labeled-test validity (↑) | APPS Avg. | TACO Avg. |
| Majority Voting | 37.88% | 43.67 | 32.67 |
| **LLM Arbiter (Ours)** | **79.88%** | **49.67** | **37.33** |

## 5. Conclusion

In this paper, we presented ADVERMCTS, a novel framework that addresses pseudo-correctness in code generation by shifting the paradigm from static verification to adversarial purification. By coupling a Solver MCTS with an active Attacker MCTS, ADVERMCTS constructs a progressively hostile environment that co-evolves with the solution candidates. This minimax interaction allows the system to autonomously discover corner cases and mitigate survivorship bias without human-annotated test oracles. Extensive experiments demonstrate that ADVERMCTS outperforms state-of-the-art search-based methods, validating the effectiveness of adversarial test-time compute. We believe this work opens new avenues for enhancing LLM reasoning robustness through autonomous, adversarial self-improvement.

## Acknowledgements

This SJTU team was supported by National Natural Science Foundation of China (62322603).

## Impact Statement

This paper presents a method for improving the robustness and correctness of automated code generation systems. Our work contributes to the reliability of AI-assisted programming, potentially reducing software bugs and development costs. However, as with any advanced code generation technology, there is a potential risk of misuse for generating malicious software. We believe the benefits of robust verification outweigh these risks, as our "Attacker" agent focuses on logical correctness rather than security exploitation. There are no specific societal consequences that we feel must be highlighted here.

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

## A. Introduction of Monte Carlo Tree Search

Monte Carlo Tree Search (MCTS) is a heuristic search algorithm that balances exploration and exploitation to navigate complex decision spaces, achieving significant success in sequential decision-making tasks (Silver et al., 2016; Shao et al., 2025; Han et al., 2024; Wang, 2025). MCTS iteratively builds a search tree where nodes represent states and edges represent actions. The process generally consists of four phases: *Selection* (select a child node that maximize a tree policy), *Expansion* (one or more child nodes are added to represent potential future states), *Simulation* (simulate until a terminal state and get reward), and *Backpropagation* (the reward is propagated back up the tree to the root and update the values of all traversed nodes). In the context of reasoning or generation tasks, MCTS allows the model to look ahead and evaluate partial solutions (Chaffin et al., 2022; Wu et al., 2025), guiding the generation process toward higher-quality outcomes compared to greedy decoding methods.

## B. Details of Baselines

To evaluate the effectiveness of ADVERMCTS, we compare it against a comprehensive set of baselines ranging from direct prompting to advanced tree-search methods. The specific implementation details for each baseline are as follows:

- **Direct Synthesis (Base & Best-of-N):** The *Base* method employs standard zero-shot prompting to generate a single solution. *Best-of-N* scales this by sampling $N = 16$ independent candidates ($T = 0.7$) and selecting the best one based on public test cases, serving as a strong sampling-based baseline without lookahead search.

- **PG-TD (Zhang et al., 2023):** This method effectively utilizes MCTS to perform lookahead search at the token level. It explores the probability space of initial tokens to identify promising prefixes, which are then deterministically completed into full code solutions, aiming to guide generation through high-likelihood token trajectories.

- **Tree of Thoughts (ToT) (Yao et al., 2023):** We implement a structured, two-layer search tree adapted for code generation. The first layer expands natural language *plans* or algorithmic sketches, while the second layer generates multiple concrete *code implementations* for each plan. This hierarchical approach decouples high-level reasoning from low-level implementation details.

- **Language Agent Tree Search (LATS) (Zhou et al., 2023):** LATS unifies reasoning and planning by searching directly in the *code space* with an integrated reflection mechanism. Upon encountering a failure during simulation, the agent generates verbal self-reflection based on the error feedback; this reflection is incorporated into the context to guide the value estimation and selection of subsequent nodes.

- **MCTS-Thought:** This is a search baseline that performs MCTS over Chain-of-Thought (CoT) reasoning steps rather than raw code. In the simulation phase, it synthesizes full code based on the current thought trajectory; crucially, execution feedback from these generated codes (e.g., test results) is added to the context to inform the expansion of the next reasoning step in a multi-turn manner.

- **RethinkMCTS (Li et al., 2025e):** Similar to MCTS-Thought, this method searches in the thought space but introduces a specific *repair mechanism*. Instead of simply discarding nodes that lead to incorrect code, it utilizes fine-grained execution feedback to explicitly "rethink" and refine erroneous thought nodes, allowing the search to recover from early reasoning mistakes without restarting.

## C. Additional Experiments

### C.1. Empirical Validation of Pseudo-Correctness: Is the Solution Generated but Overlooked?

**Motivation.**   A core premise of ADVERMCTS is that existing search methods (e.g., MCTS-Thought) suffer from pseudo-correctness: they successfully generate correct solutions during the search process, but fail to identify them due to the sparsity of public test cases (typically $|\mathcal{T}_{\text{pub}}| = 5$. To validate this hypothesis, we conducted a controlled experiment to decouple "generation capability" from "verification capability".

**Experimental Setup.**   We employed the MCTS-Thought baseline with the Qwen3-4B-Instruct model. Instead of changing the search algorithm, we only varied the verification environment used during the selection and backpropagation phases. We compared three settings:

- **Original (5 Tests):** Standard setting with 5 public tests.

- **Half Tests (50%):** Using 50% of the hidden test suite as visible constraints.

- **Oracle (All Tests):** Using the full hidden test suite (simulate perfect verification).

**Results and Analysis.** As shown in Figure 8, the performance of the exact same search algorithm improves dramatically as the verification environment becomes stricter.

- On APPS, increasing test coverage boosts Pass@1 from 43.44% to 55.33% (+11.89%).

- On TACO, the gain is even more pronounced, jumping from 31.00% to 43.67% (+12.67%).

These results provide compelling evidence that **pseudo-correctness is the primary bottleneck**. The search space already contains robust solutions (as evidenced by the high performance in the Oracle setting), but the standard sparse filter allows fragile solutions to "survive" and crowd them out. This justifies the design of ADVERMCTS: since we cannot access the Oracle tests in practice, we must actively generate adversarial tests to approximate this strict verification environment.

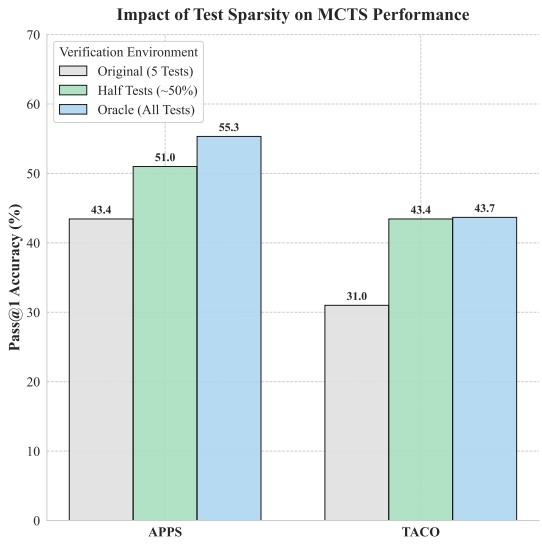

*Figure 8.* **Empirical Validation of Pseudo-Correctness.** Comparison of MCTS-Thought performance under varying verification environments (Original, Half-Hidden, Oracle). The significant performance gap between the standard setting (5 Tests) and the Oracle setting confirms the prevalence of pseudo-correctness: robust solutions are successfully generated but are filtered out due to the sparsity of public tests.

### C.2. Hard Re-ranking Outperforms Soft Penalties.

We investigate the optimal integration of adversarial feedback by comparing *Direct Penalization* (continuous value subtraction) with our *Test-based Re-ranking* (discrete filtering). Table 4 demonstrates that re-ranking consistently yields superior performance. This result indicates that adversarial tests are most effective when utilized as hard constraints rather than soft regularizers. While direct penalization introduces noise due to arbitrary scaling (e.g., equating core logic failures with minor edge cases), re-ranking aligns with the binary nature of unit testing, strictly filtering brittle solutions that fail to survive the hostile environment.

### C.3. Impact of Attacker Search Budget: More is Not Always Better.

A natural question is whether allocating more test-time compute to the Attacker (i.e., more rollouts) monotonically improves the Solver by producing increasingly challenging adversarial tests. To study this, we vary the Attacker rollout limit from 1 to 4 while keeping other components fixed, and evaluate downstream Solver performance on APPS.

Figure 9 reveals a clear inverted-U behavior: increasing rollouts from 1 to 2 yields a substantial gain (e.g., Pass@1 on APPS-Competition improves from 31% to 36%), indicating that a minimal search budget is necessary for MCTS to move beyond

*Table 4.* Signal Utilization Strategy. Comparison of using adversarial signals as a continuous penalty (*Direct Penalty*) versus a discrete filter (*Re-ranking*). The *Re-ranking* strategy consistently outperforms direct penalization, suggesting that adversarial tests are most effective when treated as hard constraints.

| Method | APPS (Pass@1) | | | TACO (Pass@1) | | |
|---|---|---|---|---|---|---|
| | Intro. | Inter. | Comp. | Easy | Medium | Hard |
| Direct Penalty | 57.0 | 45.0 | **36.0** | 49.0 | 32.0 | 17.0 |
| Re-ranking | **61.0** | **52.0** | **36.0** | **55.0** | **37.0** | **20.0** |
| *Improvement* | *+4.0* | *+7.0* | *+0.0* | *+6.0* | *+5.0* | *+3.0* |

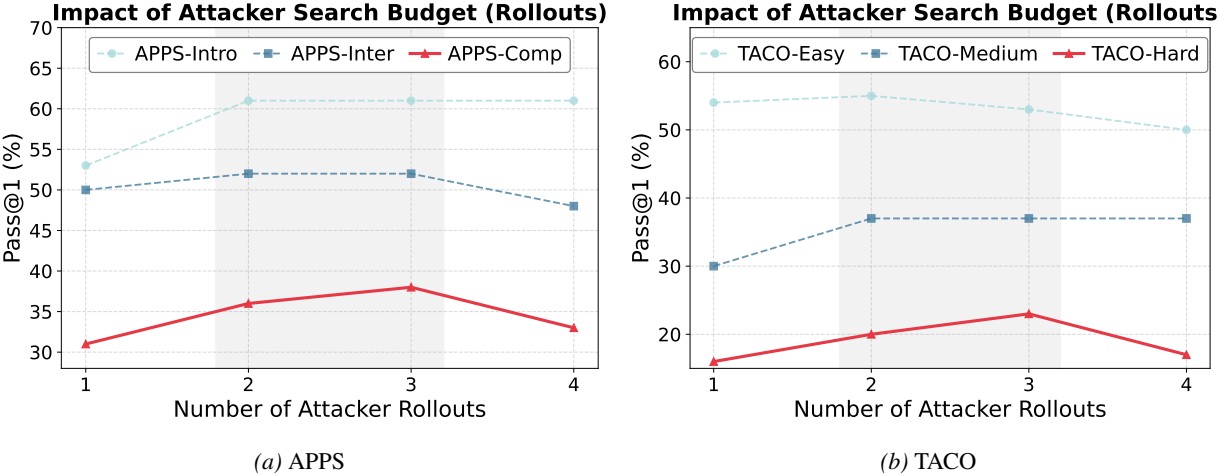

*(a)* APPS      *(b)* TACO

*Figure 9.* Attacker rollout scaling. Varying the Attacker's number of rollouts shows an inverted-U trend on both APPS and TACO: moderate budgets improve Pass@1 across difficulty splits, while larger budgets can saturate or degrade performance.

trivial cases and discover discriminative corner tests. However, further increasing the budget brings diminishing returns and can even hurt performance. We attribute this to a practical trade-off: a more aggressive search may over-optimize the divergence objective and produce tests that are less reliable as supervision signals (e.g., brittle constraints or borderline-valid corner cases), which can distort screening and re-ranking. Based on this observation, we use a moderate rollout budget ($N = 2$) in ADVERMCTS as it provides a favorable balance between effectiveness and efficiency.

## D. Prompts

In this section, we present the prompts used when an LLM acts as an agent to perform various operations.

### D.1. Solver Expansion Prompt

We present the prompt we use to instruct the LLM to sample code generation thoughts based on previous thoughts and possible previous code execution feedback.

---

**Prompt D.1: Solver Expansion Prompt**

```
SYSTEM:
You are a code reasoning agent.
Your task is to generate one plan step of reasoning for solving the programming
problem.
Strictly output ONLY the thought content without any prefix or formatting.
Be concise but comprehensive (2-3 sentences maximum).

USER PROMPT:
[Problem]
[Previous Thoughts]
[Previous Code Implementation]
```

```
[Previous Execution Results]
Provide the next logical thought to solve the problem.  If previous thoughts exist,
build upon them to offer deeper insight.
Ensure the reasoning leads to code that handles edge cases and prevents the current
error from recurring.
```

## D.2. Solver Code Generation Prompt

We present the prompt used to instruct the LLM to generate code based on previous thoughts.

**Prompt D.2: Solver Code Generation Prompt**

**SYSTEM:**
```
You are a code generator.
```
**USER PROMPT:**
```
[Problem Description]
[Previous Thoughts]
Based on the above problem description and the reasoning process (thoughts), please
generate the complete Python code to solve the problem.
The solution should contain the complete program including all the imports.
Generate the code ONLY. No other explanation or words attached!
Please wrap the solution into '''python ...  ''' format.
```

## D.3. Attacker Expansion Prompt

We present the prompt we use to instruct the LLM to sample test generation thoughts based on previous thoughts and possible previous test execution feedback on the code pool.

**Prompt D.3: Attacker Expansion Prompt**

**SYSTEM:**
```
You are a test thought designer specializing in finding bugs through systematic
testing approaches.
Your task is to propose a distinct testing thought that could reveal bugs in code
implementations.
Output your strategy directly.
Keep the thought concise (1-2 sentences) and actionable.
```
**USER PROMPT:**
```
[Problem Description]
[Previous Thoughts]
You should propose a MORE SPECIFIC sub-thought that refines the current approach.
Previous Execution Feedback:
[feedback context]
Generate a distinct testing thought that could help find bugs or differences
between code implementations.
Focus on:
1.  Edge cases and boundary conditions
2.  Special input patterns
3.  Common algorithmic pitfalls for this type of problem
4.  Input formats that often cause subtle bugs
DO NOT generate specific test inputs - only describe the STRATEGY/APPROACH.
Your thought:
```

## D.4. Attacker Test Input Generation Prompt

We present the prompt we use to instruct the LLM to generate test input following previous thoughts and based on current codes in the code pool.

**Prompt D.4: Attacker Test Input Generation Prompt**

```
SYSTEM:
You are a helpful AI Assistant that provides well-reasoned and detailed
responses.  You first think about the reasoning process as an internal monologue
and then provide the user with the answer.  Respond in the following format:
<think>...</think><answer>...</answer>.  Think concisely in 2-3 sentences.
─────────────────────────────────────────────────────────────────────────
USER PROMPT:
[Problem Description]
[Previous Thoughts]
Current Code Implementations to Test:
[codes]
Based on the testing thought above, generate ONE specific test input that:
1.  Follows the thought approach
2.  Is valid according to the problem description
3.  Is likely to reveal differences between the code implementations OR expose bugs
The codes have already passed the following public tests (use this context to find
DIFFERENT corner cases):
[public tests]
Format your answer as a JSON object containing only the "input" key, enclosed in
triple backticks ```json ```.  For example:
<answer>
```json
{"input":  "[your generated test case input]"}
```
</answer>
Important:  Only provide the INPUT, not the expected output.  We will execute the
codes to compare their outputs.
```

## D.5. Attacker Test Output Arbiter Prompt

We present the prompt we use to instruct the LLM to determine the correct output from the diverse outputs from the code pool.

**Prompt D.5: Attacker Test Output Arbiter Prompt**

```
USER PROMPT:
You are an expert programmer.  Given a programming problem and test input, multiple
code implementations produced different outputs.  Determine which output is
CORRECT.
Problem Description:
[Problem Description]
Public Tests (Use as reference for correct behavior):
[public tests]
Test Input:
[Test Input]
Code Outputs:
[Outputs]
Instructions:
1.  Understand what the problem asks
2.  Trace through the logic with the given test input
3.  Determine the CORRECT output and which code(s) produced it
Respond in the following format:
<reasoning>
Brief explanation (2-3 sentences max) of why this is the correct output.
</reasoning>
<correct_output>
The correct output value
</correct_output>
<correct_codes_id>
List of correct code indices, e.g., [1, 3] or [2]
```

```
</correct_codes_id>
```

# E. Algorithm

We present the detailed procedure of ADVERMCTS in pseudocode in Algorithm 1.

---

**Algorithm 1** The **ADVERMCTS** Inference-Time Search Procedure.

---

**Require:** $P$: Problem description; $\mathcal{T}_{pub}$: Public tests; $N_{s\_iter}$: Iterations of Solver; $N_{a\_iter}$: Iterations of Attacker.
1: **Output:** Robust solution $C^*$
2: # Initialization
3: Initialize Solver Tree $\mathcal{S}$ with root $s_0 = \{P\}$
4: Initialize Attacker Tree $\mathcal{A}$ with root $\sigma_0 = \{P, \emptyset\}$
5: Code Pool $\mathcal{C}_{pool} \leftarrow \emptyset$, Global Test Filter $\mathcal{T}_{global} \leftarrow \emptyset$
6: **for** $i \leftarrow 1$ **to** $N_{s\_iter}$ **do**
7:     # Phase 1: Solver Tree (Code Generation)
8:     $s_{leaf} \leftarrow$ SELECT($\mathcal{S}$)
9:     $s_{new} \leftarrow$ EXPAND($s_{leaf}$)
10:     $C \leftarrow$ GENERATE_CODE($s_{new}$)
11:     $r_{pub} \leftarrow$ EVALUATE($C, \mathcal{T}_{pub}$)
12:     # Strict Admission
13:     **if** $r_{pub} = 1.0$ **and** PASS_ALL($C, \mathcal{T}_{global}$) **then**
14:         $\mathcal{C}_{pool} \leftarrow \mathcal{C}_{pool} \cup \{C\}$
15:         **if** $|\mathcal{C}_{pool}| > K_{max}$ **then**
16:             Remove oldest/lowest-score code from $\mathcal{C}_{pool}$
17:         **end if**
18:     **end if**
19:     # Phase 2: Attacker Tree (Vulnerability Discovery)
20:     **if** $|\mathcal{C}_{pool}| \geq 2$ **then**
21:         **for** $i \leftarrow 1$ **to** $N_{a\_iter}$ **do**
22:             $\sigma_{leaf} \leftarrow$ SELECT($\mathcal{A}$)
23:             # Late-Binding: Thought + Code Pool $\rightarrow$ Concrete Test
24:             $T_{candidates} \leftarrow$ GEN_TEST($\sigma_{leaf}, \mathcal{C}_{pool}$)
25:             $t^* \leftarrow \arg\max_{t \in T_{candidates}}$ DIV_SCORE($t, \mathcal{C}_{pool}$)
26:             $d^* \leftarrow$ DIV_SCORE($t^*, \mathcal{C}_{pool}$)
27:             **if** $d^* > \theta_{thresh}$ **then**
28:                 # Arbiter determines Ground Truth
29:                 $o^*, \mathcal{C}_{fail} \leftarrow$ ARBITER($P, t^*,$ OUTPUTS($\mathcal{C}_{pool}, t^*$))
30:                 **if** $o^*$ is valid **then**
31:                     $\mathcal{T}_{global} \leftarrow \mathcal{T}_{global} \cup \{(t^*, o^*)\}$
32:                     **for all** $C \in \mathcal{C}_{fail}$ **do**
33:                         Apply Penalty $-V$ to Solver nodes of $C$
34:                         Update Code Value $Q(s) \leftarrow Q(s) - V$
35:                     **end for**
36:                 **end if**
37:             **end if**
38:             EXPAND($\sigma_{leaf}$)
39:             BACKPROP($\mathcal{A}, d^*$)
40:         **end for**
41:     **end if**
42:     # Phase 3: Solver Update
43:     BACKPROP($\mathcal{S}, r_{pub} -$ AccumulatedPenalties)
44: **end for**
45: # Inference: Hierarchical Test-based Reranking
46: Final Candidates $\mathcal{C}_{final} \leftarrow$ GET_ALL_CODES($\mathcal{S}$)
47: **return** $\arg\max_{C \in \mathcal{C}_{final}} \langle$PASS($C, \mathcal{T}_{pub}$), PASS($C, \mathcal{T}_{global}$)$\rangle$

---

