# OpenReview forum: "AdverMCTS: Combating Pseudo-Correctness in Code Generation via Adversarial Monte Carlo Tree Search"
_ICML.cc/2026/Conference — ICML 2026 regular_

### Official Review · Reviewer_5B67 · 2026-02-25

**Soundness:** 3
**Presentation:** 3
**Significance:** 3
**Originality:** 3
**Overall Recommendation:** 4
**Confidence:** 3

**Summary:**

This paper targets pseudo-correctness in LLM-based code generation: candidates that pass a small set of public tests but fail hidden tests due to overfitting to a weak, static verification signal. The authors propose AdverMCTS, which reframes test-time code generation as a minimax-style game between (1) a Solver that performs MCTS over chain-of-thought steps to synthesize candidate programs and (2) an Attacker that performs MCTS to generate targeted corner-case tests conditioned on the evolving pool of public-test–passing code. When the attacker finds an input that induces output divergence among code candidates, an LLM-based arbiter determines the correct output; validated tests are accumulated in a global adversarial test filter that is used both to screen future candidates (before adding them to the code pool) and to re-rank final solutions. Experiments on APPS and TACO show improved hidden-test performance versus prior search baselines, along with ablations suggesting each module contributes to gains.

**Compliance With Llm Reviewing Policy:**

Affirmed.

**Key Questions For Authors:**

# Key Questions for Authors

1. Arbiter reliability: How often does the LLM arbiter produce incorrect labels on generated adversarial tests? Can you characterize typical arbiter failure modes, and how sensitive overall performance is to them?

2. Validation/ambiguity criteria: What exact rules/prompts are used to decide that an attacker-generated input is “ambiguous or invalid” and should be discarded? How robust is performance to changing this acceptance threshold/heuristic?

3. Code pool dynamics: How does the size limit and eviction strategy for the code pool affect attacker effectiveness? Have you tried diversity-promoting strategies and do they matter?

**Limitations:**

The limitations discussion could be strengthened by the following aspects:

1. Explicitly analyzing arbiter induced error propagation (adding wrong constraints)

2. More direct measurement of compute/latency costs and scalability in practice

3. Discuss how the method behaves when divergence signals are weak or when the attacker over-optimizes toward brittle/edge validity tests.

**Strengths And Weaknesses:**

# Strengths

1. Clear problem framing: the paper identifies a realistic and common failure mode of execution. The controlled “stronger verification environment improves the same search algorithm” analysis supports the premise that verification, not generation, is often the bottleneck.


2. The overall mechanism is coherent: attacker-generated tests become hard constraints via a global filter, and the solver is forced to produce candidates robust to an increasingly hostile test suite.

# Weaknesses

The practical compute/latency overhead may be nontrivial in real deployment (dual-agent search + execution + arbiter calls), and the paper could better quantify wall-clock impacts or provide clearer guidance for compute-constrained settings.


The approach’s effectiveness may depend on having enough diversity in the initial code pool; if the solver quickly collapses to similar solutions, the attacker may have fewer divergence opportunities.

---

> ### Author Rebuttal · Authors · 2026-03-31
>
> Thank you for dedicating your time and effort to our work.
>
> > Compute / latency overhead and guidance for compute-constrained settings.
>
> We would like to respond from two aspects:
> * **Test-Time Scaling Context:** The primary application of our algorithm—and test time scaling methods in general—is tackling exceptionally challenging algorithmic problems where standard decoding fails. In these extreme scenarios, achieving robust correctness takes precedence over minimizing raw inference cost.
> * **Compute-Constrained Settings:** The framework is highly adjustable. For strict latency budgets, appropriately reducing the search rollout budget or Code Pool size provides a favorable cost-performance trade-off without catastrophic degradation.
>
> > The approach’s effectiveness may depend on having enough diversity in the initial code pool.
>
> The code pool keeps the diversity by:
> * **Strategy-Level Search:** To prevent the Solver from collapsing into highly similar implementations, our MCTS operates over the *thought and reasoning* space (strategies) rather than the raw code token space.
> * **Inherent Diversity:** Searching at the strategy level inherently yields significantly higher structural and logical diversity [1], preventing the Solver from collapsing into similar implementations and ensuring the Attacker always has sufficient discrepancies to exploit.
>
> > Arbiter reliability: How often does the LLM arbiter produce incorrect labels on generated adversarial tests? Can you characterize typical arbiter failure modes, and how sensitive overall performance is to them?
>
> * **High Empirical Reliability:** As shown in **Table 3** in the paper, the LLM Arbiter achieves a Labeled-test validity of 79.88%, which is a substantial improvement over the 37.88% validity of Majority Voting.
> * **Mechanism and Failure Modes:** The Arbiter adjudicates by comparing the divergent outputs produced by the code pool against the problem description. A typical failure mode only occurs when a flawed candidate produces an output that deceptively aligns better with the problem's surface logic than the genuinely correct output. Generating such plausible "distractors" is difficult, keeping the false-positive rate low. We will add a qualitative error analysis in the revision.
>
> > Validation/ambiguity criteria: What exact rules/prompts are used to decide that an attacker-generated input is “ambiguous or invalid” and should be discarded? How robust is performance to changing this acceptance threshold/heuristic?
>
> We apologize for omitting the exact arbiter prompt in the appendix; we will correct in the revision.
> * **Adjudication Rules:** The Arbiter is instructed to select the correct output from the diverse outputs produced by the code pool on a generated test input. In rare cases where it cannot confidently determine which output is correct, the test is deemed ambiguous and discarded.
> * **Prompt Structure:** We use the following prompt to enforce strict reasoning before concluding the correct output:
>
> ```text
> You are an expert programmer. Given a programming problem and test input, multiple code implementations produced different outputs. Determine which output is CORRECT.
>
> ## Problem Description:
> {problem}
>
> ## Public Tests (Use as reference for correct behavior):
> {public_tests}
>
> ## Test Input:
> {test_input}
>
> ## Code Outputs:
> {outputs_text}
>
> ## Instructions:
> 1. Understand what the problem asks
> 2. Trace through the logic with the given test input
> 3. Determine the CORRECT output and which code(s) produced it
>
> Respond in the following format:
> <reasoning>
> Brief explanation (2-3 sentences max) of why this is the correct output.
> </reasoning>
> <correct_output>
> The correct output value
> </correct_output>
> <correct_codes_id>
> List of correct code indices, e.g., [1, 3] or [2]
> </correct_codes_id>
> ```
>
> > Code pool dynamics: How does the size limit and eviction strategy for the code pool affect attacker effectiveness? Have you tried diversity-promoting strategies and do they matter?
>
> To directly address the impact of Code Pool capacity on the Attacker's effectiveness, we conducted additional experiments evaluating maximum Code Pool sizes of 2, 4, and 8. We will include these results in the appendix:
>
> *Impact of Code Pool Size on Pass@1 (Avg %)*
> | Dataset | Size=2 | Size=4 | Size=8|
> | :--- | :--- | :--- |:--- |
> |APPS|**49.67**|46.33|47.67|
> |TACO|**37.33**|35.33|34.67|
>
> Counter-intuitively, Size = 2 consistently achieves the highest Pass@1. An excessively large pool introduces redundant or overly complex divergent outputs, which confuses the LLM Arbiter and increases discarded tests. A smaller, tightly evicted pool ensures the Attacker focuses exclusively on the most critical logical discrepancies between top candidates, maximizing filtering efficacy.
>
> [1] Zhang, W., etc. Nl-debugging: Exploiting natural language as an intermediate representation for code debugging. EMNLP 2025.

---

> > ### Author Rebuttal · Reviewer_5B67 · 2026-04-03
> >
> > Thanks for the clarification. I don't have any further questions.

---

> > > ### Author Response · Authors · 2026-04-03
> > >
> > > Thank you very much for reviewing our rebuttal and for confirming that all your concerns have been fully resolved. We are truly glad that our clarifications were helpful!
> > >
> > > Since you kindly noted that there are no further questions, we respectfully ask if you might consider updating your score to reflect the resolved concerns.
> > >
> > > Thank you again for your time, constructive feedback, and support throughout the review process!

---

### Official Review · Reviewer_9Qcr · 2026-03-04

**Soundness:** 3
**Presentation:** 3
**Significance:** 2
**Originality:** 2
**Overall Recommendation:** 4
**Confidence:** 4

**Summary:**

The paper proposes AdverMCTS, a framework designed to reduce pseudo-correctness in LLM-based code generation, i.e., situations where generated programs pass the visible public tests but fail hidden evaluation tests. The key idea is to frame code generation as an adversarial search process between two agents: a Solver, which uses Monte Carlo Tree Search (MCTS) to generate candidate programs through structured reasoning, and an Attacker, which also uses MCTS to generate adversarial test cases that expose logical flaws in those programs. A global test filter accumulates discovered corner cases and uses them to penalize fragile solutions and refine the search process. By iteratively co-evolving programs and tests, the system builds a progressively stronger verification environment that helps distinguish truly correct solutions from ones that overfit the public tests. Experiments on competitive programming benchmarks (APPS and TACO) show that AdverMCTS improves pass rates and pass@1 accuracy over prior search-based code generation methods.

**Compliance With Llm Reviewing Policy:**

Affirmed.

**Final Justification:**

The rebuttal has addressed most of my concerns.

**Key Questions For Authors:**

1. Can the authors clarify the practical inference cost of AdverMCTS compared to simpler baselines?
2. How does AdverMCTS differ conceptually and empirically from ATGen?
3. Can the authors clarify the difference between Pass Rate and Pass@1?
4. How well does AdverMCTS scale to larger frontier models and real-world software engineering tasks?

**Limitations:**

Please take a look at the weaknesses pointed out in the review, and either address them in the rebuttal or incorporate them as limitations of the paper.

**Strengths And Weaknesses:**

Strengths:
1. **Interesting problem formulation.** The paper identifies pseudo-correctness (i.e., solutions that pass public tests but fail hidden tests) as an important challenge in code generation and proposes an adversarial formulation to address it.

2. **Empirical improvements on APPS and TACO.** The proposed method consistently outperforms several search-based baselines on competitive programming benchmarks.

3. **Useful ablation studies.** The paper evaluates the impact of several core components (e.g., attacker tree, multi-sample test generation, filtering hub), helping justify the design choices.

Weaknesses:
1. **Heavy reliance on heuristic components.** Several key parts of the system (e.g., divergence-based rewards, LLM arbiter, filtering strategy) are largely heuristic, and the paper provides limited discussion of their reliability or theoretical justification.

2. **Limited discussion of computational cost.** The method introduces a dual-agent MCTS framework with additional test generation and verification steps, but the practical inference cost and runtime overhead are not thoroughly analyzed, especially compared to based decoding that is widely used in practice.

3. **Relation to ATGen is unclear.** The paper does not clearly discuss the conceptual or empirical differences between the proposed approach and ATGen.

4. **Benchmarks remain narrow.** Evaluation focuses exclusively on competitive programming datasets (APPS and TACO), leaving it unclear whether the method generalizes to real-world software engineering tasks.

5. **Limited improvement over the strongest baseline.** While the method outperforms prior approaches, the gains over the strongest baseline (e.g., RethinkMCTS) appear relatively modest in several settings.

6. **Evaluation metrics are not sufficiently explained.** The distinction between Pass Rate and Pass@1 is not clearly discussed, which makes it harder to interpret the reported improvements.

7. **Experiments focus mainly on relatively small models.** Most experiments use smaller open models, and it is unclear how well the approach scales to larger frontier models commonly used in practice.

---

> ### Author Rebuttal · Authors · 2026-03-31
>
> Thank you for dedicating your effort to our work.
>
> > Heavy reliance on heuristic components.
>
> We respectfully clarify that these are not ad-hoc heuristics, but direct consequences of our core idea: **if weak public tests cause pseudo-correctness, then test-time scaling must strengthen verification, not just expand search.** Accordingly, the main designs follow naturally: **divergence** is the signal of under-tested behaviors, the **arbiter** is needed to resolve which divergent behavior is more likely correct, and **the global filter** is needed to accumulate such evidence so fragile solutions are consistently excluded.
>
> These modules are thus tightly coupled parts of a single adversarial verification loop, not independent tricks. We will clarify this derivation in the revision, and the ablation supports it: removing AT-Tree, DMTS, or GF-Hub each consistently lowers Pass@1.
> > Limited discussion of computational cost.
>
> We provide a cost-performance Pareto analysis in Figure 5. Our method achieves a better Pareto frontier than standard rollout scaling, indicating that the extra compute is spent on effective verification rather than redundant sampling. More broadly, test-time scaling is intended for challenging algorithmic problems where standard decoding fails, so robust correctness is the primary objective rather than minimizing raw inference cost.
> > Distinction from ATGen
>
> While both works involve adversarial ideas, AdverMCTS is fundamentally different from ATGen in both goal and method design:
>
> * **Stage & Objective**: ATGen is a **training-time** Reinforcement Learning (RL) framework designed to train a standalone test generator. Its primary goal is to address reward sparsity and the lack of difficult buggy code during the training phase of a test generation model. In contrast, AdverMCTS is an **inference-time** search algorithm specifically designed to combat the pseudo-correctness of generated codes caused by sparse public tests.
> * **Method Design**: In ATGen, adversarial code mainly serves as data augmentation for updating the test generator. In AdverMCTS, the Attacker and Solver interact online during search: the Attacker conditions on the current code pool, searches for divergence-inducing tests, and these validated tests are immediately used to filter and rerank candidates.
>
> > Benchmarks remain narrow... whether the method generalizes to real-world software engineering tasks.
>
> We focus on APPS and TACO to strictly isolate core reasoning capabilities. We will clarify this design choice and discuss broader applications in the revision:
>
> * **Isolating Reasoning:** Competitive programming provides a pure testbed that isolates a model's deep logical deduction and complex algorithmic reasoning. Real-world software engineering tasks involve additional factors such as long-context repository understanding, retrieval, and tool interaction, which would introduce confounding variables beyond our current focus.
> * **Generalizing to SWE:** This does not imply **the idea** is limited to competitive programming. Rather, the underlying principle of using adversarially discovered counterexamples to filter pseudo-correct candidates may also be useful in broader software engineering settings.
>
> > Limited improvement over the strongest baseline.
>
> We respectfully disagree that the improvement is marginal. Our main metric, Pass@1, is a strict problem-level metric: a gain here means more problems are solved completely on the hidden test suite, rather than merely improving partial execution quality. Thus, even a few points correspond to a meaningful number of previously unsolved problems becoming fully solved. On APPS with Qwen3-4B-Instruct, for example, AdverMCTS improves over RethinkMCTS from 43.67 to 49.67 Pass@1; on TACO with the same backbone, it improves from 33.67 to 38 on average. We will make this interpretation clearer in the revision.
> > Evaluation metrics are not sufficiently explained.
>
> We will update to clearly distinguish the two metrics:
> * **Pass Rate:** The average percentage of hidden test cases successfully passed by the generated code. This metric reflects the *partial correctness* of the solutions.
> * **Pass@1:** A strict, binary evaluation indicating whether the single, final selected code candidate successfully passes *all* hidden test cases for a given problem. This represents absolute, robust correctness.
>
> > Experiments on frontier model.
>
> We would like to respond in two aspects:
> * We respectfully direct your attention to **Figure 3** and Section 5.3, where we demonstrate that AdverMCTS maintains a significant lead and successfully pushes the performance limits on the state-of-the-art **DeepSeek-V3.2 (671B)** frontier model.
> * We utilized the smaller Qwen3 models for our extensive experiments primarily for computational feasibility and because it already possesses top-tier reasoning capabilities for its size, making it an ideal and representative open-weight testbed.

---

> > ### Author Rebuttal · Reviewer_9Qcr · 2026-04-04
> >
> > Thank you for the rebuttal. It has resolved most of my questions, and I have increased the review score accordingly. However, I still believe the generalization of the proposed approach to SWE tasks is an important direction, and it is worth adding a discussion about this. I hope the other review points could also be incorporated into the paper.

---

> > > ### Author Response · Authors · 2026-04-04
> > >
> > > Thank you so much for taking the time to read our rebuttal, and for increasing your score. We deeply appreciate your constructive feedback and support!
> > >
> > > We agree with your insight that generalizing our proposed approach to broader Software Engineering (SWE) tasks is a promising direction. We will certainly add a dedicated discussion exploring this potential adaptation in the final manuscript.
> > >
> > > Furthermore, we will ensure that all other clarifications discussed during the rebuttal phase are carefully and comprehensively incorporated into the paper.
> > >
> > > Thank you again for your time and valuable guidance!

---

### Official Review · Reviewer_J711 · 2026-03-13

**Soundness:** 3
**Presentation:** 3
**Significance:** 3
**Originality:** 3
**Overall Recommendation:** 4
**Confidence:** 3

**Summary:**

This paper addresses the problem of enhancing code generation tasks by augmenting the standard Monte Carlo Tree Search with a new component, AdverMCTS. During the tree search process, the framework identifies new testing strategies and generates additional test cases to expand the test pool. This approach improves code generation performance, with results demonstrating that the method is both efficient and outperforms current baselines.

**Compliance With Llm Reviewing Policy:**

Affirmed.

**Final Justification:**

The rebuttal addresses my concerns about discussing new test cases and cost analysis with Rethink-MCTS. I will maintain my score.

**Key Questions For Authors:**

1. The experiments are based on the Qwen series of models (some with Deepseek). I am not asking for an exhaustive evaluation across different model families, but I am wondering how performance would change for a different model family. Additionally, how this method would work on larger models? Does AdverMCTS scale efficiently to even larger models?

**Limitations:**

yes

**Strengths And Weaknesses:**

**Strength**
- The paper is well written and easy to follow.
- The experiment is thorough, and the results show good performance.

**Weakness**
- I'm wondering what kind of new test cases AdverMCTS can generate. I do hope the author can provide more detailed analysis so that we can know what exactly is needed from the static code pool. I believe such explanation can further strengthen the paper.
- The second-best performer is Rethink-MCTS. Since the current cost analysis only compares to MCTS-Thought and the gain over Rethink-MCTS appears marginal, I believe the cost analysis should specifically include a comparison to Rethink-MCTS.
- How does AdverMCTS compare to search-based code generation methods, such as [1]?

[1] AlphaEvolve: A coding agent for scientific and algorithmic discovery

---

> ### Author Rebuttal · Authors · 2026-03-31
>
> Thank you for dedicating your time and effort to our work.
>
> > what kind of new test cases AdverMCTS can generate... what exactly is needed from the static code pool.
>
> We appreciate the opportunity to clarify the interactive relationship between the generated tests and the Code Pool.
>
> **1. What kind of new test cases AdverMCTS generates:**
> The Attacker does not generate generic or random extra test cases. Instead, it synthesizes targeted adversarial corner cases. These inputs are designed to induce execution divergences among the highly plausible candidate codes.
>
> **2. What is needed from the Code Pool:**
> To make the Attacker's targeted generation effective, the Code Pool contains candidates that might exhibit **pseudo-correctness**—specifically, codes that have already successfully passed all public tests.
> * **The Baseline Problem:** Sparse public tests cannot distinguish genuinely correct solutions from fragile, over-fitted ones.
> * **The Sieve Mechanism:** The Code Pool serves as a concentrated set of these highly deceptive candidates. By conditioning on this pool, the Attacker analyzes the subtle logical discrepancies among them and instantiates concrete inputs to force them into outputting differently.
> * **Distilling Correctness:** The generated tests act as a dynamic sieve. By iteratively applying these adversarial tests to penalize and filter out the codes that fail, the framework systematically eliminates the "pseudo-correct" implementations. Ultimately, the robustly correct code is the only one that survives this rigorous purification process.
>
> > Cost analysis with Rethink-MCTS.
>
> We appreciate the suggestion. We originally selected MCTS-Thought for the cost-performance Pareto frontier because it serves as the most token-efficient search baseline. While RethinkMCTS is effective, its repair mechanism relies heavily on verbose, fine-grained execution feedback, which substantially inflates token consumption.
>
> To address your concern, we evaluated the token efficiency between AdverMCTS and RethinkMCTS. Due to space limits, we present the comparison on the APPS dataset below:
>
> **Performance vs. Token Cost (Pass@1 % / Avg Tokens per problem)**
>
> | Method | Intro | Inter | Comp |
> | :--- | :--- | :--- | :--- |
> | RethinkMCTS | 56.0 / 191.7k | 47.0 / 194.1k | 29.0 / 254.6k |
> | **AdverMCTS** | **61.0 / 176.7k** | **52.0 / 180.4k** | **36.0 / 224.5k** |
>
> As shown, AdverMCTS achieves significantly higher robust correctness (Pass@1) while actually consuming **fewer** tokens than RethinkMCTS.
>
> Furthermore, we want to emphasize the broader context of test-time scaling:
> * **Primary Objective:** These long-thinking frameworks are designed to tackle exceptionally challenging algorithmic problems where standard decoding completely fails.
> * **Value of Compute:** In such extreme scenarios, unlocking the capability to solve previously unsolvable problems (robust correctness) takes absolute precedence over minimizing raw inference token cost.
>
> > How does AdverMCTS compare to search-based code generation methods, such as AlphaEvolve?
>
> Our paper already compares against **representative search-based code generation methods**, including LATS, MCTS-Thought, and Rethink-MCTS. These are the most directly relevant baselines for our setting because they address competition-level code generation, where the main challenge is solving algorithmic programming problems under sparse verification.
>
> Regarding AlphaEvolve, we agree it is a valuable related work, but its primary setting is different from ours. AlphaEvolve is framed as a coding agent for scientific and algorithmic discovery, whereas our focus is on competition-level code generation as a controlled testbed for studying reasoning robustness and pseudo-correctness. The core capability stressed in our setting is not open-ended scientific discovery, but finding correct algorithmic solutions despite weak public-test supervision.
> > Generalization beyond Qwen and scalability to larger models
>
> We respectfully direct your attention to **Figure 3** and Section 5.3 ("Scaling with Backbone Capability"), where we specifically addressed scaling.
>
> * **Choice of Qwen3:** We utilized the Qwen3 series (4B and 8B) for our main experiments because their advanced reasoning capabilities and open-weights nature facilitated extensive, cost-effective ablation studies.
> * **Evaluation on DeepSeek-V3.2:** To verify model-agnosticism, we evaluated AdverMCTS on the state-of-the-art frontier model, DeepSeek-V3.2 (671B), via its official API.
> * **Efficient Scaling:** As shown in Figure 3 in the paper, the results demonstrate a strictly monotonic upward trend across model scales. AdverMCTS maintains a significant lead over strong baselines and scales highly efficiently, confirming that adversarial test-time compute pushes the boundaries even on massive frontier models.

---

> > ### Author Rebuttal · Reviewer_J711 · 2026-04-01
> >
> > Thanks to the authors for the additional experiments and comments.
> >
> > Regarding the first weakness, I understand that the attacker synthesizes adversarial corner cases. What I mean for "what kind of new test cases AdverMCTS can generate" is that could the authors provide some concrete examples showing the normal cases from the pool and the new generated cases?
> >
> > Regarding AlphaEvolve, I think while AlphaEvolve is designed for scientific discovery, it is still conceptually related at a higher level because it performs iterative code evolution guided by evaluator feedback. Therefore, even if it is not the most direct experimental baseline, I believe it is still worth discussing explicitly in the paper for clearer positioning.

---

> > > ### Author Response · Authors · 2026-04-01
> > >
> > > Thank you for your continued engagement and the constructive follow-up.
> > >
> > > > Concrete Example: How AdverMCTS Exposes "Silent Bugs"
> > >
> > > To illustrate the difference between normal cases and AdverMCTS-generated cases, let us look at a specific problem from the APPS dataset (Problem 29: "Lucky Ticket").
> > >
> > > **Problem**: Given a 6-digit string, find the minimum number of digit replacements required to make the sum of the first three digits equal to the sum of the last three digits.
> > >
> > > Public Tests:
> > >
> > > - `000000 -> 0`
> > > - `123456 -> 2`
> > > - `111000 -> 1`
> > > - Why they are insufficient: None of these public tests require more than 2 changes to balance the ticket. The maximum sum difference tested is 9 (for 123456).
> > >
> > > **The Pseudo-Correct Trap:**
> > > Conditioned heavily on these simple public tests, the Solver frequently generates "pseudo-correct" solutions utilizing flawed, lazy logic. For example, one surviving implementation in our pool correctly calculates the initial sums and checks if 0 or 1 replacements are sufficient. However, if 1 change is not enough, it bypasses any further algorithmic logic and hardcodes a default output of 2 (else: print(2)), completely ignoring the possibility that 3 changes might be required.
> > >
> > > Why it survives: This code perfectly passes the public tests because the public suite simply does not contain any extreme edge cases that demand 3 changes. Standard search methods (like MCTS-Thought) cannot detect this hardcoded "silent bug" and confidently select this flawed code.
> > >
> > > **The AdverMCTS Generated Cases (Adversarial Tests):**
> > > Instead of generating generic random strings, the Attacker analyzes the implementations in the code pool and actively synthesizes corner cases that exploit these rigid directional heuristics. We present one concrete adversarial test generated here:
> > >
> > > - `123999` (A massive sum difference of 21)
> > >
> > > When the generated test 123999 is fed into the pool, a divergence occurs:
> > > - The pseudo-correct code fails to recognize the extreme difference (which requires 3 changes, as a single replacement can at most shift the sum by 9) and lazily triggers its hardcoded fallback, incorrectly outputting 2.
> > > - A genuinely robust code in the pool (which properly enumerates combinations up to 3 changes rather than relying on a hardcoded fallback) correctly outputs 3.
> > >
> > > This is the core advantage of AdverMCTS: with only one test case generation, it can locate the specific logic flaw in the code. It does not just generate "more" tests; it dynamically searches for the exact logical boundaries (like extreme sum differences requiring maximum replacements) that the "surviving" pseudo-correct codes have failed to handle. We will add a qualitative case study section in the Appendix of our revision to include this exact example.
> > >
> > > Besides the example above, AdverMCTS also routinely generates structurally complex test cases to expose intricate algorithmic flaws (e.g., subtle errors in dynamic programming state transitions or graph traversals). Because the execution traces of these advanced adversarial cases are too lengthy to unpack here, we may include more complex adversarial scenarios in the Appendix of our revision.
> > >
> > > > Discussion on AlphaEvolve
> > >
> > > We agree with your assessment regarding AlphaEvolve. While its primary testbed is open-ended scientific and algorithmic discovery, the high-level conceptual mechanism—iterative code evolution guided by evaluator feedback—is relevant to our framework.
> > >
> > > We will explicitly include a discussion of AlphaEvolve in the Related Work section of our revision. We will use it to provide clearer positioning, highlighting how iterative evaluator feedback can be adapted from open-ended scientific discovery (AlphaEvolve) to the specific challenge of mitigating pseudo-correctness under strict, test-sparse competitive programming environments (AdverMCTS).
> > >
> > > ****
> > > As the conference policy permits only a single round of author-reviewer discussion during this phase, we sincerely hope this concrete example and our planned revisions address your remaining questions. If our response has resolved your concerns, we would be deeply grateful if you might consider reflecting this by raising your score.

---

### Official Review · Reviewer_pmSp · 2026-03-13

**Soundness:** 3
**Presentation:** 3
**Significance:** 3
**Originality:** 3
**Overall Recommendation:** 5
**Confidence:** 4

**Summary:**

This paper proposes AdverMCTS, an inference-time scaling method to improve LLM code generation performance and reduce pseudo-correctness problems where LLM fails on hidden tests. It introduces an adversarial MCTS process to generate corner test cases and select the best code candidates and improve the robustness of generated code. Evaluations demonstrate superior performance over MCTS baselines and ablations have shown the importance of the adversarial MCTS process.

**Compliance With Llm Reviewing Policy:**

Affirmed.

**Final Justification:**

The rebuttal addressed most of my concerns. While I still have concern on the data contamination and encourage the author to evaluate LiveCodeBench or other benchmarks that could rule out the data contamination risks (e.g. avoiding the framework to have memorized ground truth tests), I think this paper has genuine contributions towards test-time scaling on LLM for coding. Therefore I raise my score to 5.

**Key Questions For Authors:**

See weakness.

**Minor**

1/ What is the “Pass Rate” in Table 1? Is it Pass@16 given you have 16 rollouts?

**Limitations:**

There is no explicit limitation discussion in the paper. I encourage the authors to discuss them, for example the dependence on an imperfect LLM arbiter, the divergence blind spot when wrong programs agree, added inference-time cost, and its failure modes.

**Strengths And Weaknesses:**

**Strengths**

1/ The paper addresses a real and important failure mode in competition-level code generation. The core method is technically coherent: a Solver MCTS searches for candidate solutions while an Attacker MCTS searches for discriminative tests, and the discovered adversarial tests are reused through a global filter and final reranking.

2/ Experimental results have shown meaningful and consistent improvements on APPS and TACO benchmarks across Qwen3 4B/8B and DeepSeek-V3.2 models.

3/ Ablations and comparison to baselines have shown the merit of the proposed framework.

4/ The presentation is easy to follow and the evaluation result is clear and with details.

**Weakness**

1/ Incomplete baseline comparison: The paper does not compare against some very relevant methods despite discussed in related works and there is no explanation why these methods are not compared, especially S* and CodeTree. For example, S* also leverages discriminative tests to select the best code candidates, while the search process is rather simpler, so it could be a great baseline to compare the plain attacker vs. the MCTS attacker. Please consider adding explanations of why these baselines are not relevant or repetitive, or compare with them in evaluation.

2/ Data contamination concern: The paper chose APPS and TACO as the core benchmarks, which were released before the evaluated LLMs. It is possible that these LLMs have seen the hidden tests and the framework is only finding a way to generate memorized tests. Evaluating on evolving and/or new benchmarks like LiveCodeBench on newer versions could further improve the validity of paper claims. I feel at least a discussion of the data contamination problem is needed.

3/ Limited model evaluation: Current evaluation is limited on open-weights models and mainly the smaller ones. It is unclear how the framework could improve frontier models including those closed-source ones. Showing improvements on frontier models could further improve the paper contributions.

4/ Soundness: While the LLM arbiter has shown significant improvement over majority voting, there is still a 20% gap. A discussion on how the errors from LLM arbiter can be identified, controlled and mitigated can further improve the soundness of the paper.

5/ Scopes: The paper focuses on competition-level coding problems. While this is still an important problem to address and the paper has stated the scope very clearly which I appreciate, I believe adding some discussions on how this framework could potentially improve broader software engineering and agentic coding methods could position this paper in a more timely and important area as well.

---

> ### Author Rebuttal · Authors · 2026-03-31
>
> Thank you for dedicating your time and effort to our work.
>
> > Incomplete baseline comparison: S* and CodeTree.
>
> We have evaluated AdverMCTS against S\* and CodeTree on both Qwen3-4B-Instruct and Qwen3-8B. We only show the averaged Pass@1 due to limited space here. As shown in the table below, AdverMCTS consistently outperforms both baselines across datasets, and we will include these results in the revision.
>
> **Performance Comparison (Pass@1 %)**
>
> | Backbone & Dataset | S* | CodeTree | **AdverMCTS** |
> | :--- | :--- | :--- | :--- |
> | **Qwen3-4B (APPS)** | 36.33 | 41.33 | **49.67** |
> | **Qwen3-4B (TACO)** | 27.00 | 29.67 | **38.00** |
> | **Qwen3-8B (APPS)** | 21.67 | 29.33 | **36.00** |
> | **Qwen3-8B (TACO)** | 15.00 | 23.00 | **28.00** |
>
> The performance delta highlights the fundamental algorithmic differences between our approach and the baselines:
> * **AdverMCTS vs. S\*:** While S\* leverages prompts to generate test inputs, it essentially creates a static filter (acting similarly to an expanded public test suite). AdverMCTS employs a dynamic, co-evolutionary process where the attacker uses MCTS to actively search for *divergence-inducing* tests based on the evolving code pool. This reward-driven selection process provides a significantly stricter verification environment than S\*'s static generation, translating to higher Pass@1 robust correctness.
> * **AdverMCTS vs. CodeTree:** CodeTree, similar to the LATS baseline, relies on a search and self-reflection mechanism (searching strategies before code). While effective, single-agent reflection struggles to identify silent edge cases. AdverMCTS’s dual-agent adversarial interaction proactively exposes vulnerabilities that a reflective solver might overlook, yielding a superior advantage, particularly on the most challenging splits (e.g., APPS-Competition).
>
>
> > Data contamination concern.
>
> We agree contamination is a ubiquitous challenge. However, our evaluation isolates the *search algorithm's* impact. If gains were purely from memorization, the Base and Base@16 performances would be much higher. The substantial delta between direct synthesis and AdverMCTS proves our dual-agent search drives the improvement. We will discuss LiveCodeBench for future un-contaminated evaluation. We appreciate the suggestion regarding LiveCodeBench and will discuss this as a vital direction for future un-contaminated evaluation.
>
> > Frontier model evaluation.
>
> Please see **Figure 3 and Section 5.3**, where we explicitly evaluated AdverMCTS on the state-of-the-art **DeepSeek-V3.2 (671B)** via API. The strictly monotonic upward trend proves our framework scales effectively to massive frontier models. We prioritized open-weights models for extensive ablations due to API cost/rate limits.
>
> > Soundness: About Errors from the LLM arbiter.
>
> We do not claim a perfect arbiter, but rather a superior signal compared to majority voting (validity improved from 37.88% to 79.88%, Table 3). About the typical failure modes and mitigations, the arbiter is only invoked when outputs already diverge, invalid tests are discarded, and the discovered tests are used mainly as filtering / reranking constraints rather than as a fully trusted oracle for end-to-end supervision. More tricks such as self-consistency, or multi-judge agreement might lead to further improve reliability but is out of scope of this paper.
>
> > Scopes and Limitations.
>
> We appreciate the suggestions to expand the paper's scope and discuss boundaries explicitly:
> * **Possible Broader Applications:** We will add a discussion detailing how the AdverMCTS framework can be adapted for broader software engineering tasks. Our core idea—using adversarially discovered counterexamples to iteratively filter pseudo-correct solutions—may also be useful in broader software engineering settings where superficial success signals can be misleading.
> * **Limitations Section:** An explicit section discussing the framework's dependence on the LLM arbiter's reasoning limits, divergence blind spots (e.g., when wrong programs agree on an output), and the fact that current experiments focus on competition-level code generation rather than broader software-engineering tasks..
>
> >  Minor: What is the “Pass Rate” in Table 1? Is it Pass@16 given you have 16 rollouts?
>
> We will update to clearly distinguish the two metrics:
> * **Pass Rate:** The average percentage of hidden test cases successfully passed by the generated code. This metric reflects the *partial correctness* of the solutions.
> * **Pass@1:** A strict, binary evaluation indicating whether the single, final selected code candidate successfully passes *all* hidden test cases for a given problem. This represents absolute, robust correctness.
>
> We report Pass@1 instead of Pass@16 because it's only allowed to select 1 final code to test on hidden test cases for all search algorithms, although the algorithm would search for 16 rollous of codes.

---

> > ### Author Rebuttal · Reviewer_pmSp · 2026-04-03
> >
> > The rebuttal addressed most of my concerns. While I still have concern on the data contamination and encourage the author to evaluate LiveCodeBench or other benchmarks that could rule out the data contamination risks (e.g. avoiding the framework to have memorized ground truth tests), I think this paper has genuine contributions towards test-time scaling on LLM for coding. Therefore I raise my score to 5.

---

### Decision · Program_Chairs · 2026-04-30

**Decision:**

Accept (regular)

**Comment:**

This paper proposes AdverMCTS, an adversarial test-time search framework for code generation that explicitly targets pseudo-correctness by coupling solver-side code search with attacker-side adversarial test generation. Reviewers agreed that the paper addresses an important and realistic failure mode in competition-level code generation and found the overall framework technically sound and empirically strong. The main remaining weaknesses are that evaluation remains centered on competition-style coding benchmarks, leaving broader generalization to software engineering tasks as future work, and that some concerns persist about benchmark contamination and reliance on an imperfect LLM arbiter.  Overall, I am positive on the paper. It makes a solid and timely contribution.